# Detection of dengue, malaria, and additional causes of acute febrile illness: The need for expanded testing, Bayelsa State, Nigeria

Dayo Olufemi Akanbi[1,2,3]*, Bio Belu Abaye[1,2], Francisco Averhoff[4,5], Michael G. Berg[4,5], Gregory S. Orf[4,5], Kabiru M. Lawan[6], Geoff A. Beckett[7], Ayotunde Bolatito Omotoso[8], Maximillian Mata[4,5], Gavin A. Cloherty[4,5], Lucious Chabuka[9], Tulio de Oliveira[9,10], Kenneth W. Mac-Fisi[2], Azibadighi Walter[2], Inara Isaac Mark[2], George Edeki[11], Abdoulaye Sinayoko[12], Muazzam Nasrullah[13], Binafeigha Ihekerenma Justina[14], Muhammad Shakir Balogun[1,15]

1 Nigeria Field Epidemiology and Laboratory Training Program, Abuja, Nigeria, 2 Epidemiology Unit, Bayelsa State Ministry of Health, Yenagoa, State Secretariat, Yenagoa, Bayelsa State, Nigeria, 3 Nigerian Correctional Service, Minna, Niger State, Nigeria, 4 Abbott Diagnostics, Abbott Park, Illinois, United States of America, 5 Abbott Pandemic Defense Coalition, Abbott Park, Illinois, United States of America, 6 Department of Veterinary Public Health and Preventive Medicine, Ahmadu Bello University, Zaria, Nigeria, 7 Independent Researcher, Decatur, Georgia, United States of America, 8 Department of Behavioural Sciences, University of Ilorin, Ilorin, Nigeria, 9 Centre for Epidemic Response and Innovation (CERI), School of Data Science and Computational Thinking, Stellenbosch University, Stellenbosch, South Africa, 10 KwaZulu-Natal Research Innovation and Sequencing Platform (KRISP), University of KwaZulu-Natal, Durban, South Africa, 11 Department of Statistics, University of Ibadan, Ibadan, Oyo State, Nigeria, 12 World Health Organization, Abuja, FCT, Nigeria, 13 Injury Control Research Center, West Virginia University, Morgantown, West Virginia, United States of America, 14 Diete-Koki Memorial Hospital Opolo, Yenagoa, Bayelsa State, Nigeria, 15 African Field Epidemiology Network, Asokoro, Abuja, Nigeria

* dayodfa@gmail.com

## Abstract

Dengue virus (DENV) infection has not been previously reported from Bayelsa State, Nigeria. We aimed to determine the prevalence of dengue virus (DENV) infection, malaria, and coinfection, and other pathogens among febrile patients in the capital city, Yenagoa.We conducted a cross-sectional study among persons aged ≥1 year who presented with acute febrile illnesses (AFI) at four hospitals in Bayelsa State during 20 May – 15 June 2022. Blood samples from 443 participants were tested for DENV seromarkers (NS1, IgM, IgG), using serology and RT-PCR, and malaria was diagnosed by thick smear microscopy. Sociodemographic and risk factor data were collected using electronic questionnaires administered via smart phones/tablets and analyzed using univariate and multivariate methods. Metagenomic libraries were prepared and enriched by viral target capture and sequenced by NGS. The seroprevalence of acute DENV infection was 14.5% (n = 64) while the prevalence of malaria was 42.4% (n = 188); 6.5% (n = 29) of participants were coinfected with acute DENV infection and malaria. An additional 17.6% (n = 78) of participants had markers for past DENV infection. Rural/suburban residence and age ≥ 31 years were significantly correlated with having any dengue seromarker. Residence in a larger household (≥5

**Data availability statement:** All relevant data are within the manuscript and its Supporting information.

**Funding:** DOA received a grant for this study from Abbott Laboratories and administered by Training Programs in Epidemiology and Public Health Interventions Network (TEPHINET), a program of the Task Force for Global Health, Inc., under contract number 3928/6424. The funders had no role in study design, data collection and analysis, decision to publish, or preparation of the manuscript.

**Competing interests:** I have read the journal's policy, and the authors of this manuscript have the following competing interests: FA, MGB, GSO, MM and GAC receive salary and may own stock in Abbott Diagnostics. The other authors have declared that no competing interests exist.

persons), and borehole water-use were predictors for malaria fever. RT-PCR results revealed multiple DENV serotypes, with serotype 3 dominant. Sequencing of unknown AFI cases revealed numerous viral causes such as adenovirus, EBV, and hepatitis A, as well as additional dengue and malarial infections missed by conventional testing. Of interest were Coxsackievirus A5 (hand, foot and mouth disease; HFMD) which has been diversifying locally for years in Nigeria and an mPox clade IIb (lineage A.2.3) strain that emerged in Nigeria during the 2022 global outbreak. The results of this study provide the first documentation of human DENV infection in Bayelsa State, Nigeria and suggests that dengue is an emerging and important cause of febrile illness in this area. Our findings support the need for routine testing to identify DENV among patients who present with acute febrile disease. Metagenomic NGS results highlight the benefits of unbiased surveillance to identify circulating and emerging pathogens.

## Author summary

In Bayelsa State, Nigeria, dengue virus (DENV) had never been confirmed as a cause of illness before now. We wanted to find out how many people with fever were infected with dengue, malaria, or both at the same time. To do this, we studied people of all ages who came to four hospitals in Yenagoa, the state capital, with a fever between May and June 2022. We took blood samples and tested them for evidence of dengue and malaria. Out of 443 people with fever, about 1 in 7 had a recent dengue infection, and more than 4 in 10 had malaria; nearly half of acute dengue infected individuals were co-infected with malaria. We also found that older people and those living in rural or suburban areas were more likely to test positive for dengue, while living in crowded homes and using borehole water were linked to malaria. We discovered that more than one type of dengue virus was present, with type-3 being the most common. The remaining undiagnosed cases were sequenced with a virus target enrichment approach and revealed a wide variety of other pathogens causing AFI. This study is the first to demonstrate that dengue virus is circulating (extensively) in Bayelsa. Our results suggest that doctors should consider testing for dengue when treating patients with fever, and that public health efforts need to address dengue as an emerging cause of AFI in the state. NGS identified a sizable proportion of additional pathogens among patients that tested negative for DENV and malaria.

## Introduction

Mosquito-borne diseases caused by dengue virus (DENV) and malaria parasites are responsible for significant morbidity and mortality globally [1, 2]. Transmission to humans occurs primarily through the bite of either infected *Aedes or Anopheles* mosquitoes for dengue and malaria fever respectively [3]. These diseases are known to contribute

significantly to the burden of human disease globally [4–6]. While malaria is routinely tested for and diagnosed, testing for DENV is not generally available in Nigeria [6, 7], and febrile patients presenting for care are usually assumed to have malaria and treated accordingly [8], often without testing [5]. The primary aim of this study was to determine the prevalence of dengue and malaria, and their predictors among febrile patients seeking care at four selected surveillance sites in Yenagoa, Bayelsa state, Nigeria, where dengue had not been previously identified. In addition, next-generation sequencing (NGS) was applied to a subset of dengue and malaria positives as well 'negatives', to identify other pathogens that may contribute to AFI in Bayelsa state.

## Methods

### Ethics statement

Prior ethical approval was received for this study from the Bayelsa State Ministry of Health Ethical Review Committee (Approval Number: BSHREC/Vol 1/21/12/02) and ethical clearance was received from the two tertiary health facilities involved in this study. Written consent was also obtained from participants and assent of children was obtained from parents/guardian of minors. Ethical approval for this study was obtained from The Bayelsa State Ministry of Health Research and Ethical Committee, Approval Number: BSHREC/Vol 1/21/12/02, and all procedures were conducted in accordance with the Declaration of Helsinki.

The study was conducted in Yenagoa, Bayelsa, the capital of the state, located in the Niger River Delta (as shown in Fig 1). Bayelsa is characterized by an equatorial climate with distinct arid and rainy seasons, an environment that is predominantly rainforest and includes freshwater wetlands which are prone to sporadic flooding. It is also a conducive habitat to arboviral disease vectors [9]. For our analysis, the study area was classified into urban, suburban and rural areas. Urban areas are defined by relatively high population density, extensive built environment (infrastructure, housing, roads), and access to services such as clean water, sanitation, and waste management. Suburban Settlements are transitional zones between urban centres and rural areas, typically characterized by moderate density, mixed residential development, and partial access to urban infrastructure. These areas in Bayelsa often include peripheral neighbourhoods of Yenagoa with growing housing and environmental challenges. Rural areas are geographic areas outside of towns and cities, with low population density, agricultural or forest land use, and limited infrastructure.

We established surveillance for acute febrile illness (AFI) at four hospitals in Yenagoa; Federal Medical Centre, Yenagoa, Diete-Koki Memorial Hospital, Niger-Delta University Teaching Hospital and Bayelsa Specialist Hospital, Yenagoa (as shown in Fig 1). AFI is a condition characterized by the sudden onset of fever, that may or may not be accompanied by additional symptoms [4]. Patients ≥ 1 year of age presenting with a body temperature of ≥ 37.5°C, excluding those with known infections or non-infectious causes of fever were eligible to be enrolled in the study.

We recruited febrile patients who sought care at any one of our four surveillance sites between 20 of May and 15 June 2022. Five milliliters (mL) of whole blood was aseptically collected by venipuncture and transferred into an EDTA specimen bottle. The sample was divided into three equal aliquots, one of which was centrifuged at 5000 × g for 5 minutes to extract plasma, which was stored at 4°C for dengue immunoassay and subsequently stored at -20°C for PCR testing for DENV, and a third aliquot for sequencing.

### Screening for plasmodium parasite

With an aliquot, we screened for plasmodium parasites in thick blood films, this was made using a standardized and validated technique as described by WHO Basic Malaria Microscopy Learner's Guide [10]. Plasmodium species was identified based on morphological characteristics of the trophozoites, schizonts, and gametocytes, observed on the thick smear.

### Detection for DENV seromarkers

We used 25μL of extracted plasma from each of the 443 consenting participants for the detection of dengue virus (DENV) seromarkers using an Acro Biotech Inc. Dengue Combo Test cassette [11]. This rapid chromatographic immunoassay

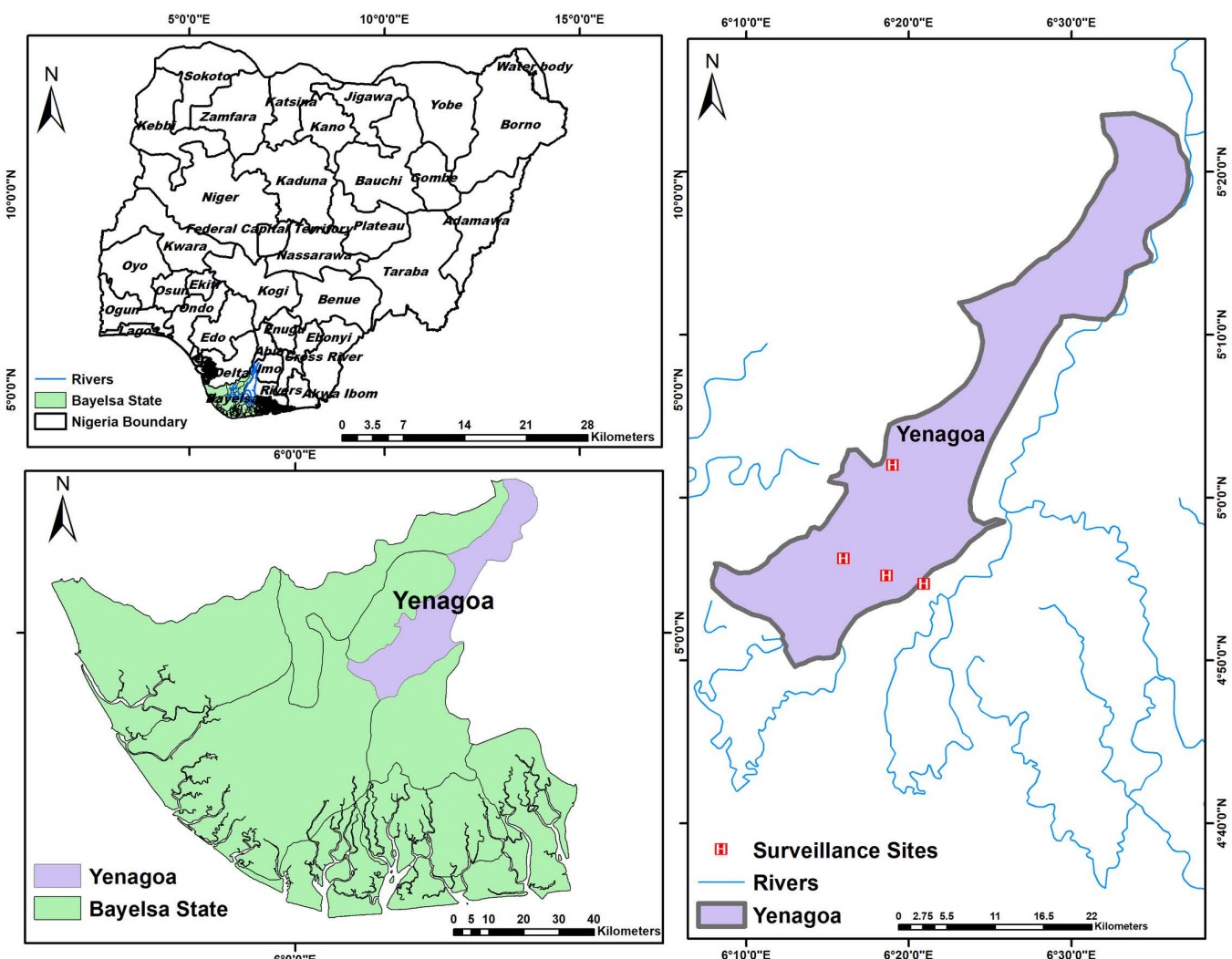

**Fig 1. Map of Nigeria showing Bayelsa state districts including Yenagoa on the bottom, left side.** On the right, red 'H' icons represent the four surveillance sites within Yenagoa LGA. Basemap sources: Administrative boundaries (Nigeria, Bayelsa State, and Yenagoa LGA): Obtained from the Office of the Surveyor General of the Federation (OSGOF), Nigeria, via the Humanitarian Data Exchange (HDX): https://data.humdata.org/dataset/nga-administrative-boundaries License: Creative Commons Attribution 4.0 International (CC BY 4.0) https://creativecommons.org/licenses/by/4.0/ Rivers shapefile: Derived from the Digital Chart of the World (DCW) dataset, available via Natural Earth: https://www.naturalearthdata.com/downloads/10m-physical-vectors/10m-rivers-lake-centerlines/ License:Public domain https://www.naturalearthdata.com/about/terms-of-use.

qualitatively detects DENV NS-1 antigen and IgG/IgM antibodies in human plasma, it also identifies primary and secondary dengue. All 443 samples were initially screened, and 60 samples testing positive for NS-1 antigen and/or IgM seromarkers were subsequently tested using RT-PCR multiplex for DENV serotyping.

## Dengue multiplex RT-PCR extraction, detection and serotyping

Nucleic acid extraction was performed using the QIAamp Viral RNA Mini Kit (Qiagen) [12], and the extracted RNA was stored at -30°C. Nucleic acid extraction is succinctly described here:

Viral RNA was extracted from plasma samples using the QIAampViral RNA Mini Kit (Qiagen), following the manufacturer's instructions. Briefly, 140 µL of plasma was mixed with 560 µL of AVL buffer containing carrier RNA. After brief centrifugation, 560 µL of ethanol was added, and the mixture was pulse-vortexed. A 630 µL portion was transferred to a QIAamp Mini spin column and centrifuged at 6000 × g. The column was washed sequentially with 500 µL of AW1 and AW2 buffers, including a high-speed spin at 20,000 × g. A final spin at 6000 × g was performed to remove residual wash buffer. RNA was eluted in 60 µL of AVE buffer and stored at –30°C for subsequent PCR analysis.

Dengue virus (DENV) serotyping was conducted using the Bosphore Dengue Virus Genotyping Kit v1 (Anatolia Geneworks) [13] on the Gentier 48E/48R Real-Time PCR System. Master Mix 1 targeted DENV-1 and DENV-4, while Master Mix 2 targeted DENV-2 and DENV-3. Each 30 µL reaction included 14.0 µL of master mix, 1.0 µL of internal control, and 15.0 µL of RNA. Thermal cycling conditions were: reverse transcription at 50°C for 30 minutes; initial denaturation at 95°C for 14 minutes 30 seconds; denaturation at 97°C for 30 seconds; annealing at 62°C for 1 minute 20 seconds; and final data collection hold at 32°C for 5 minutes. Fluorescent channels FAM, HEX, and CY5 were used to detect specific DENV serotypes. The DENV immunoassay and PCR detection kits used in this study were purchased from Inqaba Biotech West Africa Inc, Ibadan Nigeria.

We defined acute dengue infection as: NS-1 positive and/or presence of IgM (with or without IgG positive) and/or PCR positive. We defined past dengue infection as: positive for IgG and negative for all other DENV markers (NS-1, IgM, and PCR).

We conducted interviews using digital questionnaires to gather data on socio-demographic/risk factors. Statistical analysis was done using Statistical Package for Social Sciences software (SPSS) version 23, we conducted univariate analysis to describe sociodemographic characteristics and infection status. Bivariate associations between independent variables and DENV or malaria outcomes were assessed using chi-square tests. Binary logistic regression model to identify independent predictors, and statistical significance was set at $p < 0.05$.

### Next generation sequencing

Patient specimens were extracted for total nucleic acid (TNA) on PerkinElmer Chemagic 360 using CMG-1033-S kit (Revvity, Waltham, MA). Metagenomic libraries were prepared after converting TNA to double stranded cDNA, followed by fragmenting and barcoding with Nextera XT (Illumina, CA) as described [14]. Libraries (n = 411) were pooled at 3 samples per capture for hybridization with Illumina Viral Surveillance Panel (VSP; Illumina) probes, adhering to the manufacturer protocol. Target-enriched pools were diluted to 2nM and further diluted to 750pM final and loaded on Nextseq-2000 instrument (Illumina) in six successive runs of between 48 and 96 samples. FASTQ files were analyzed by DiVir 3.0, a proprietary cloud-based metagenomic pipeline. Read mapping and consensus sequence generation were performed with CLC Bio Genomics Workbench (v25) and BioEdit software.

### Phylogenetic analysis

The VP1 region of our CVA5 sequence was queried using an internal BLAST server. The accessions and regions matched through BLAST (plus metadata) were pulled from GenBank on July 25, 2025 (including three CVA12 sequences to be used as an outgroup), then aligned using the LINSI function of MAFFT. The phylogeny of this dataset was inferred using the maximum likelihood method; briefly, the alignment was processed by IQ-TREE2, using the ModelFinder module to determine the best nucleotide substitution model, and 1000 replicates of ultrafast bootstrapping were performed to provide statistical support to branches. The trees were visualized using the ggtree package in R.

### Results

We recruited 443 participants during the study period. Seroprevalence for any DENV seromarkers, past or current/acute infection was 32.1% (n = 142). The prevalence of seromarkers for acute/current infection (NS-1 and IgM) was 14.5% (n = 64) while for past DENV infection (IgG) was 17.6% (n = 78) (See Table 1). Prevalence of malaria was (n = 188) 42.4%

**Table 1. Socio-demographic Characteristics of Study Participants with Dengue and/or malaria.**

| Characteristics | DENV+(NS1+IgM) (n = 64) | Malaria (n = 188) | DENV+(IgG) (n = 78) | Co-infection (Malaria+ and DENV+) (n = 29) | Negative (n = 178) |
|---|---|---|---|---|---|
| | n (%) | n (%) | n (%) | n (%) | n (%) |
| **Sex** | | | | | |
| Male | 26 (40.6) | 72 (38.3) | 30 (6.8) | 11 (37.9) | 68(38) |
| Female | 38 (59.4) | 116 (61.7) | 48 (10.8) | 18 (62.1) | 110(62) |
| **Age (years)** | | | | | |
| 0 − 9 | 11 (17.2) | 36 (19.1) | 6 (1.4) | 4 (13.8) | 55 (31) |
| 10 − 19 | 7 (10.9) | 23 (12.2) | 3 (0.7) | 5 (17.2) | 26 (15) |
| 20 − 29 | 9 (14.1) | 23 (12.2) | 7 (1.6) | 2 (6.9) | 20 (11) |
| 30 − 39 | 12 (18.8) | 35 (18.6) | 13 (2.9) | 4 (13.8) | 34 (19) |
| 40 − 49 | 14 (21.9) | 36 (19.1) | 29 (6.5) | 7 (24.1) | 23 (13) |
| 50 − 59 | 6 (9.4) | 23 (12.2) | 11 (2.5) | 4 (13.8) | 13 (7) |
| 60 − 69 | 4 (6.3) | 10 (5.3) | 8 (1.8) | 2 (6.9) | 4 (2) |
| ≥ 70 | 1 (1.6) | 2 (1.1) | 1 (0.2) | 1 (3.4) | 3 (2) |
| **Marital Status** | | | | | |
| Single | 14 (21.9) | 41 (21.8) | 17 (3.8) | 6 (20.7) | 55 (31) |
| Married | 35 (54.7) | 90 (47.9) | 45 (10.2) | 14 (48.3) | 60 (34) |
| Co-habiting | 0 | 2 (1.1) | 1 (0.2) | 0 | 1 (1) |
| Divorced/Separated | 0 | 3 (1.6) | 4 (0.9) | 0 | 1 (1) |
| Widowed | 4 (6.3) | 4 (2.1) | 3 (0.7) | 3 (10.3) | 4 (2) |
| Child | 7 (10.9) | 40 (21.3) | 6 (1.4) | 5 (17.2) | 47 (26) |
| Others | 3 (4.7) | 5 (2.7) | 1 (0.2) | 1 (3.4) | 9 (5) |
| Undisclosed | 1 ((1.6) | 3 (1.6) | 1 (0.2) | 0 | 1 (1) |
| **Family structure** | | | | | |
| Monogamous | 43 (67.2) | 128 (68.1) | 45 (10.2) | 17 (58.6) | 113 (63) |
| Polygamous | 5 (7.8) | 17 (9.0) | 8 (1.8) | 2 (6.9) | 19 (11) |
| Undisclosed | 16 (25.0) | 43 (22.9) | 25 (5.6) | 10 (34.5) | 46 (26) |
| **Household size** | | | | | |
| < 5 | 26 (41.9) | 88 (47.8) | 37 (8.4) | 15 (53.6) | 70 (39) |
| ≥ 5 | 36 (58.1) | 96 (52.2) | 39 (8.8) | 13 (46.4) | 108 (61) |
| **Educational Level** | | | | | |
| No formal | 5 (7.8) | 11 (5.9) | 4 (0.9) | 2 (6.9) | 18 (10) |
| Elementary/Primary | 7 (10.9) | 32 (17.0) | 4 (0.9) | 2 (6.9) | 46 (26) |
| Secondary | 16 (25.0) | 51 (27.1) | 21 (4.7) | 8 (27.6) | 35 (20) |
| Tertiary | 36 (56.3) | 90 (47.9) | 46 (10.4) | 17 (58.6) | 75 (42) |
| Professional/Technical | 0 | 3 (1.6) | 2 (0.5) | 0 | 3(2) |
| Undisclosed | 0 | 1 (0.5) | 1 (0.2) | 0 | 1(1) |
| **Occupation** | | | | | |
| Civil servant | 24 (37.5) | 61 (32.4) | 35 (7.9) | 12 (41.4) | 51(29) |
| Business/Trading | 14 (21.9) | 36 (19.1) | 23 (29.5) | 5 (17.2) | 23(13) |
| Farming | 1 (1.6) | 2 (1.1) | 1 (0.2) | 1 (3.4) | 6(3) |
| Student | 18 (28.1) | 50 (26.6) | 9 (2.0) | 8 (27.6) | 58(33) |
| Unemployed | 1 (1.6) | 8 (4.3) | 2 (0.5) | 0 | 11(6) |
| Others | 5 (7.8) | 29 (15.4) | 7 (1.6) | 2 (6.9) | 28(16) |
| Undisclosed | 1 (1.6) | 2 (1.1) | 1 (0.2) | 1 (3.4) | 1(1) |

*(Continued)*

**Table 1.** (Continued)

| Characteristics | DENV+(NS1+IgM) (n = 64) | Malaria (n = 188) | DENV+(IgG) (n = 78) | Co-infection (Malaria+ and DENV+) (n = 29) | Negative (n = 178) |
|---|---|---|---|---|---|
| | n (%) | n (%) | n (%) | n (%) | n (%) |
| **Residence** | | | | | |
| Rural | 5 (7.8) | 21 (11.2) | 13 (2.9) | 1 (3.4) | 19(11) |
| Sub-urban | 43 (67.2) | 107 (56.9) | 44 (9.9) | 22 (75.9) | 97(54) |
| Urban | 16 (25.0) | 59 (31.4) | 21 (4.7) | 6 (20.7) | 62(35) |
| Undisclosed | 0 | 1 (0.5) | 0 | 0 | 0 |

(all malaria positives were *Plasmodium falciparum*), and the co-infection rate with acute/current DENV infection was 6.5% (n = 29). Participants ≥31 years of age were less likely to test positive for any DENV seromarker than those <31 years of age (aOR: 0.38, 95% CI: 0.24–0.60). In addition, persons living in rural/suburban areas were twice as likely to test positive for any marker of DENV compared to residents of urban areas (aOR: 2.12, 95% CI:1.02–4.40). Having a household size >5 residents (aOR: 1.52, 95% CI:1.03–2.24) and using a borehole as water source (aOR: 1.77, CI:1.02–3.06)) were associated with increased risk of malaria. Of the 23 acute/current DENV positive samples tested using RT-PCR serotyping test for DENV, serotype 1 accounted for 4.3% (n = 1), serotype-2 accounted for 8.7% (n = 2), and serotype-3 accounted for 87.0% (n = 20). We also observed co-infection of serotypes in two of these patients, there was also a concurrence of two or more seromarkers in some patients as shown in Table 2.

Next-generation sequencing was performed on 411/443 participants to assess concordance with existing dengue and malaria data, as well as to explore undiagnosed causes of AFI. There were n = 46 dengue IgM positives, of which n = 16 (35%) were PCR positive. Only 11/16 were included for sequencing, and with 10/11 having a Ct ≥ 28, only sample #400 (K059054) with Ct = 26 (also NS1+) was positive by NGS. However, three IgM+ samples testing negative by PCR (1/3 was NS1+) were weakly positive by NGS. From the n = 17 NS1 antigen positive samples, only 6/17 were PCR+, and of the 14/17 sequenced, only 2 were dengue NGS+; 1 of 2 was PCR+. NGS of undiagnosed individuals detected an additional n = 3 dengue positives who were negative by rapid test and PCR; two were genotype 1 and one was genotype 3. Sample K05911, a genotype 1 with 99% nucleotide identity to a strain also collected in Nigeria in 2021 (OR259174.1) obtained 71% genome coverage at 10X depth (Fig 2A).

Despite using virus probe enrichment (teNGS), we have observed that non-viral pathogens may bind non-specifically and still be detected by sequencing. Of the n = 156 thick blood smear positives sequenced by NGS, only n = 20 were also positive for malaria by NGS (Fig 2B), although negative results were generally in agreement (n = 183). As teNGS is not intended to capture *P. falciparum*, it is therefore not surprising that n = 136 smear-positives were not detected, but it was notable that n = 36 smear-negatives were identified by NGS, including one *P. ovale* and one *P. malariae* infection.

Several other viral infections were detected, including hepatitis A, parvovirus B19, herpesviruses (CMV, EBV), and respiratory viruses such as adenovirus (species A, B, C) and SARS-CoV-2 (Fig 2C). Notably, we detected an mPox infection (K058971) which yielded 23% coverage from 2267 reads (257 RPM). NextStrain classified it as clade IIb, lineage A.2.3, as it harbored the C175759T & G170317A signature mutations. The sequence blasted to accession MT250197.1 recovered in 2019 in Singapore from a Nigerian traveler. Finally, a Coxsackievirus A5 (CVA5, also referred to as hand-foot-and mouth disease; HFMD) sequence was recovered from sample K059080 with 98% genome coverage (7266 nt) at 20X depth (1967 reads, 240 RPM). A phylogenetic analysis of the VP1 region indicates CVA5 is divided into two major lineages, one found in Europe and Asia, and another in Europe and Africa (Fig 2D, *left panel*). Our K059080 isolate (accession number: **PX056340**) found in the latter lineage (Africa/Europe; expanded *right panel*) is situated near the root of a cluster of 81 isolates from Europe and the U.S. (clade collapsed for clarity). It is likely descended from an African

**Table 2. Seromarker and serotype profile of qRT-PCR Positive Patients.**

PCR Positive patients (n = 23)

| Patient ID | IgM | NS-1 | IgG | DENV Serotype(s) |
|---|---|---|---|---|
| 1 | – | – | Positive | 1, 4 |
| 2 | – | Positive | – | 3 |
| 3 | – | Positive | – | 3 |
| 4 | Positive | – | – | 3 |
| 5 | Positive | – | – | 3 |
| 6 | Positive | – | – | 3 |
| 7 | Positive | – | – | 3 |
| 8 | Positive | – | – | 3 |
| 9 | – | Positive | – | 3 |
| 10 | – | Positive | – | 3 |
| 11 | Positive | – | – | 3 |
| 12 | Positive | – | – | 3 |
| 13 | Positive | – | – | 3 |
| 14 | Positive | Positive | Positive | 3 |
| 15 | Positive | Positive | – | 3 |
| 16 | Positive | Positive | – | 2, 3 |
| 17 | Positive | Positive | – | 3 |
| 18 | Positive | Positive | – | 1 |
| 19 | Positive | – | – | 3 |
| 20 | Positive | – | – | 3 |
| 21 | Positive | – | – | 2 |
| 22 | Positive | – | – | 3 |
| 23 | Positive | – | – | 3 |

ancestor, branching from other strains collected in Ethiopia, Madagascar, Cameroon, and Ghana. It most closely resembles another strain collected from Nigerian wastewater in 2018 (MW373942), indicating ongoing, local diversification. Overall, of the 443 patients enrolled, we found through malaria and DENV testing and NGS, at least one possible etiology of AFI for 317/443 (71%) of the patients.

## Discussion

This is the first study to our knowledge documenting DENV infection in Bayelsa State, Nigeria. We found a substantial burden of DENV infection among febrile patients seeking care in the study area, in addition to expected larger burden of malaria infection. This study also revealed a significant number of coinfections with malaria and DENV which have clinical and public health implications [5], including the potential for greater disease severity, risk of misdiagnosis, and missed opportunities for appropriate medical management. Finally, several additional causes of AFI were identified by application of NGS methods, giving a more complete understanding of the causes of AFI.

The high proportion of participants with evidence of past dengue infection suggests that unrecognized (cryptic) circulation of infection has been ongoing in Bayelsa. Coinfections among patients with AFI are consistent with entomological findings that the study area harbors vectors of transmission for both malaria and DENV, including *Aedes, Anopheles* and *Mansonia* species [9].

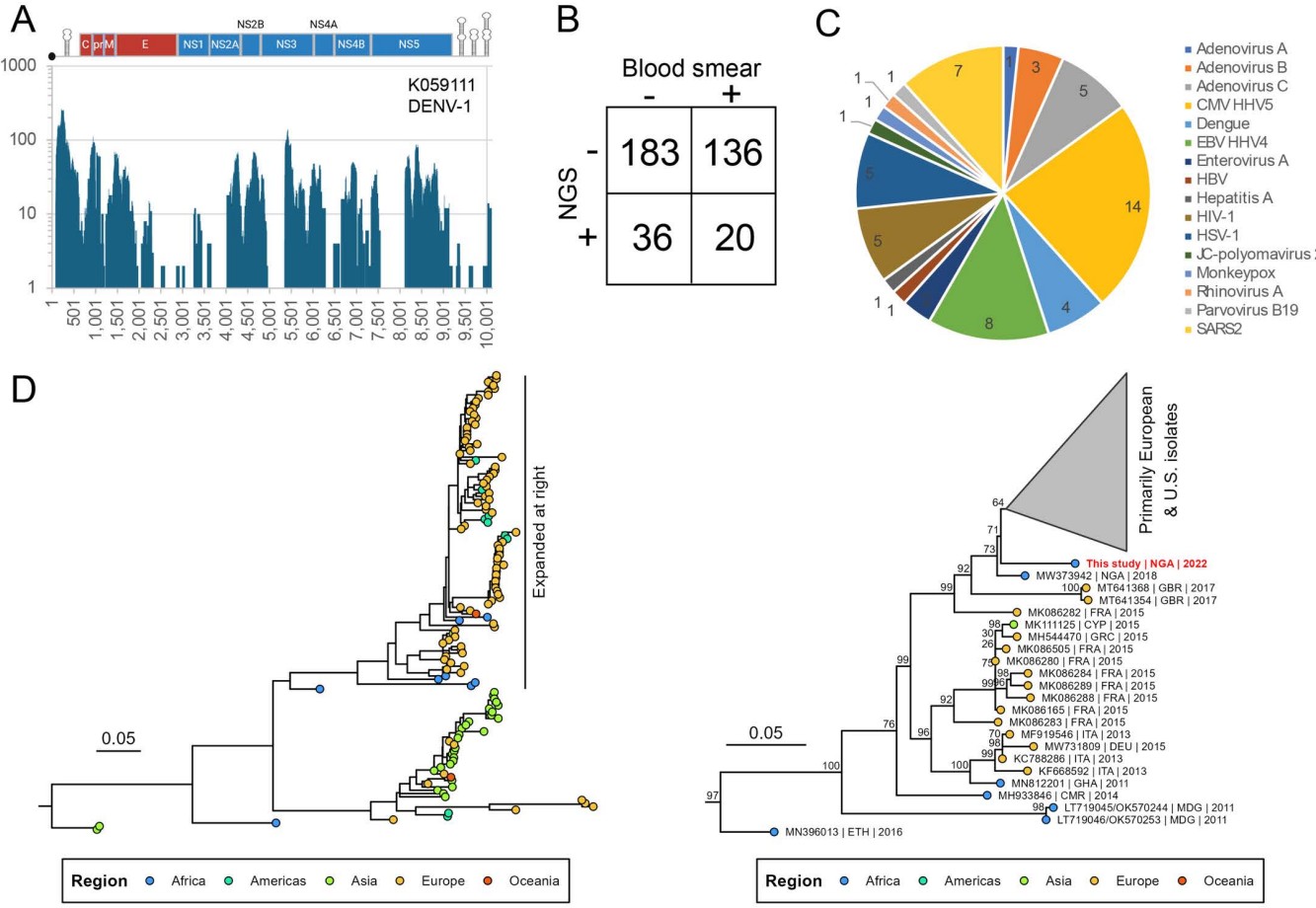

**Fig 2. Target-enriched NGS detects a variety of viral causes of acute febrile illness.** A. NGS coverage plot of DENV-1 infection. B. Malaria detection concordance by NGS and thick blood smear. C. Breakdown of viral species detected by NGS. D. Maximum likelihood phylogeny of 145 VP1 sequences from Coxsackievirus A5 (HFMD virus) rooted using Coxsackievirus A12 as an outgroup. Full phylogeny (left panel) and zoomed subset of the major African-European clade (right panel) are shown. Tips are colored by collection region labels follow the convention "accession | three-letter country code | collection year", with the sequence from this study highlighted in red.

The higher proportion of dengue cases among rural and suburban residents versus urban residents, can be attributed to a combination of factors such as vector breeding conditions, human behavior, protective infrastructure, and healthcare access which may be responsible for higher dengue burden in rural and suburban areas [15].

The variation in DENV, malaria, and co-infection rates across age groups reflects the influence of key biological, behavioural, immunological and environmental factors. However, there were no observed significant variations in clinical signs between the groups of patients with malaria, dengue, or dengue – malaria coinfection.

Among children aged 0–9 years, there was an outstanding burden of malaria and dengue, likely due to their increased immunological vulnerability, inadequate use of protective measures such as insecticide-treated nets, and behavioural patterns which may increase their exposure to dengue vectors. Those in 10–29-year age group, had infection rates which were generally lower. This may be attributed to stronger immune responses, greater awareness, and adoption of personal protective behaviours, resulting in fewer cases and co-infections. Past infection which was evidenced by high IgG seromarker was most demonstrated by those in the 30–49 age group, this may be reflective of their overall exposures over time based on their geographical locations combined with increased occupational mobility and social interactions

that raise the likelihood of contact with both Aedes and Anopheles vectors. These may also account for the higher DENV, malaria and coinfection rates. The higher acute dengue burden in this age group aligns with findings from a study in southwestern Nigeria [16]. Among individuals aged 50 years and above, infection rates were lower for acute dengue, possibly due to reduced outdoor mobility, behavioural differences, or age-related decline in immune responsiveness.

In this study, some individuals were positive for multiple DENV seromarkers, they had both NS1/IgM and IgG. Individuals with a previous DENV infection are at risk for dengue hemorrhagic fever (DHF) or dengue shock syndrome (DSS) with subsequent DENV infection [5]. With subsequent DENV infection, there is the possibility of development of antibody–antigen complexes in secondary, tertiary, or quaternary infections which may trigger the complement cascade resulting in a DHF [17]. It is therefore important to maintain vigilance for such cases in Bayelsa State.

The multiple circulating dengue serotypes revealed by molecular testing with multiplex RT-PCR suggest ongoing co-circulation in Bayelsa, which may increase the risk of outbreaks of more severe forms of dengue such as dengue hemorrhagic fever (DHF) or dengue shock syndrome (DSS). Serotyping allows for monitoring of new introductions, allowing for early warning of potential outbreaks, and may improve prevention and control of dengue epidemics and patient management [18].

Next-generation sequencing confirmed the circulation of DENV1 and DENV3 and identified additional infections missed by available diagnostics (Fig 2A). Similarly, discordant malaria reads found in specimens that were blood-smear negative for malaria might be dismissed as false positives, or it may be that NGS is more sensitive at detecting malaria than microscopy in some instances (Fig 2B). A range of viral infections were detected (n = 60), but the majority (n = 298, 72.5%) remained undiagnosed after sequencing. The monkeypox case is consistent in terms of geography and timing, with clade IIb causing the 2017–2019 outbreak in Nigeria and expanding globally in 2022 [19]. With outbreaks still actively occurring in neighboring countries, vigilance is imperative [20]. HFMD is a painful, highly contagious infection that spreads rapidly and typically affects young children. Although greater surveillance is needed to better understand its evolutionary history, it appears that this lineage has participated in numerous intercontinental transmission events.

There were several important findings from this study. First, the need to expand access to DENV testing for febrile patients. This requires educating clinicians in hospitals and enhancing surveillance across the state to monitor trends, identify outbreaks, and improve patient care in this newly identified DENV-endemic region of Nigeria.

In addition, the findings from NGS highlight the multitude of pathogens in circulation, including epidemic-prone agents like HFMD and Mpox. Finally, the possible under-diagnosis of malaria, from the specimens where it was only detected by NGS deserves additional study.

## Study limitations

A limitation of this study was the absence of comprehensive clinical and laboratory data to assess disease severity among participants with acute febrile illness. While the study successfully identified the prevalence and co-infection rates of dengue and malaria using serological, molecular, and metagenomic approaches, it lacked key clinical parameters such as complete blood counts (e.g., white blood cell and platelet levels), liver function tests (e.g., AST, ALT), hematocrit levels, and other biomarkers indicative of disease progression. This limits the ability to differentiate between mild and severe forms of dengue or to detect complications

## Supporting information

**S1 Data. MINNA_FINAL_ABRIDGED_PLOS.** Raw dataset in Excel format used for the analysis of dengue and malaria co-infections among febrile patients in Bayelsa State, Nigeria. Sheet 1: Individual level data on infection status, socio-economic and demographical characteristics.
(XLSX)

## Acknowledgments

Profound gratitude to the Almighty God for the gift of life and making this project possible. I appreciate the managements of Abbott, Training Programs in Epidemiology and Public Health Interventions Network (TEPHINET), NCDC, AFENET, Bayelsa State Ministry of Health, Federal Medical Center Yenagoa, Diete-Koki Memorial Hospital, Bayelsa Specialist Hospital and Bayelsa Dengue Project Team and patients who participated for helping to actualize this work.

## Author contributions

**Conceptualization:** Dayo Olufemi Akanbi, Bio Belu Abaye, Francisco Averhoff, Muazzam Nasrullah.

**Data curation:** Dayo Olufemi Akanbi, Francisco Averhoff, Michael G Berg, Gregory S. Orf, Azibadighi Walter, George Edeki, Abdoulaye Sinayoko.

**Formal analysis:** Michael G Berg, Gregory S. Orf, George Edeki.

**Funding acquisition:** Dayo Olufemi Akanbi.

**Investigation:** Dayo Olufemi Akanbi, Kenneth W. Mac-Fisi, Azibadighi Walter, Inara Isaac Mark.

**Methodology:** Dayo Olufemi Akanbi, Bio Belu Abaye, Francisco Averhoff, Michael G Berg, Gregory S. Orf, Kabiru M. Lawan, Maximillian Mata, Lucious Chabuka, Muazzam Nasrullah.

**Project administration:** Dayo Olufemi Akanbi, Bio Belu Abaye, Francisco Averhoff, Kabiru M. Lawan, Gavin A. Cloherty, Tulio de Oliveira, Muazzam Nasrullah, Binafeigha Ihekerenma Justina.

**Resources:** Dayo Olufemi Akanbi, Francisco Averhoff, Gavin A. Cloherty, Binafeigha Ihekerenma Justina.

**Software:** Dayo Olufemi Akanbi, Abdoulaye Sinayoko.

**Supervision:** Dayo Olufemi Akanbi, Bio Belu Abaye, Francisco Averhoff, Kabiru M. Lawan, Geoff A. Beckett, Ayotunde Bolatito Omotoso, Maximillian Mata, Tulio de Oliveira, Muhammad Shakir Balogun.

**Validation:** Bio Belu Abaye, Geoff A. Beckett, Ayotunde Bolatito Omotoso, Tulio de Oliveira, Binafeigha Ihekerenma Justina.

**Visualization:** Dayo Olufemi Akanbi, Michael G Berg.

**Writing – original draft:** Dayo Olufemi Akanbi, Francisco Averhoff, Michael G Berg, Gregory S. Orf.

**Writing – review & editing:** Dayo Olufemi Akanbi, Francisco Averhoff, Michael G Berg, Gregory S. Orf, Kabiru M. Lawan, Ayotunde Bolatito Omotoso, Muhammad Shakir Balogun.

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
