## [Decision Letter · Decision Letter 0]

7 May 2025

The need for routine testing for dengue virus in Nigeria: detection and prevalence of dengue and coinfection with malaria among febrile patients in Yenagoa, Bayelsa State, Nigeria

Dear Dr. Akanbi,

Thank you for submitting your manuscript to PLOS Neglected Tropical Diseases. After careful consideration, we feel that it has merit but does not fully meet PLOS Neglected Tropical Diseases's publication criteria as it currently stands. Therefore, we invite you to submit a revised version of the manuscript that addresses the points raised during the review process.

Please submit your revised manuscript within 60 days Jul 06 2025 11:59PM. If you will need more time than this to complete your revisions, please reply to this message or contact the journal office at plosntds@plos.org. Please include the following items when submitting your revised manuscript:

We look forward to receiving your revised manuscript.

Kind regards,

Ran Wang, M.D.

Academic Editor

Qu Cheng

Section Editor

Shaden Kamhawi

co-Editor-in-Chief

Paul Brindley

co-Editor-in-Chief

**Journal Requirements:**

At this stage, the following Authors/Authors require contributions: Geoff A. Beckett, and Azibadighi Walter. Please ensure that the full contributions of each author are acknowledged in the "Add/Edit/Remove Authors" section of our submission form.

- ® on page: 5 and 6.

4) Tables should not be uploaded as individual files. Please remove these files and include the Tables in your manuscript file as editable, cell-based objects. For more information about how to format tables, see our guidelines:

https://journals.plos.org/plosntds/s/tables 

5) We have noticed that you have uploaded Supporting Information files, but you have not included a list of legends. Please add a full list of legends for your Supporting Information files after the references list.

Potential Copyright Issues:

- Figure 1. Please (a) provide a direct link to the base layer of the map (i.e., the country or region border shape) and ensure this is also included in the figure legend; and (b) provide a link to the terms of use / license information for the base layer image or shapefile. We cannot publish proprietary or copyrighted maps (e.g. Google Maps, Mapquest) and the terms of use for your map base layer must be compatible with our CC BY 4.0 license.

7) Please amend your detailed Financial Disclosure statement. This is published with the article. It must therefore be completed in full sentences and contain the exact wording you wish to be published. Please ensure that the funders and grant numbers match between the Financial Disclosure field and the Funding Information tab in your submission form. Note that the funders must be provided in the same order in both places as well.

**Reviewers' Comments:**

Reviewer's Responses to Questions

**Key Review Criteria Required for Acceptance?**

**Methods:**

-Are the objectives of the study clearly articulated with a clear testable hypothesis stated?

-Is the study design appropriate to address the stated objectives?

-Is the population clearly described and appropriate for the hypothesis being tested?

-Is the sample size sufficient to ensure adequate power to address the hypothesis being tested?

-Were correct statistical analysis used to support conclusions?

-Are there concerns about ethical or regulatory requirements being met?

Reviewer #1: appropriate

Reviewer #2: 1. Is the Acro Biotech kit an Abbott dengue combo? The author should add a reference to Acro Biotech. Can that kit identify dengue primary or secondary infection? The reviewer thinks that the information is helpful for readers.

2. How much serum was used for RNA extraction, and how much ul of RNA was applied for real-time RT-PCR? The reviewer believes this information is helpful to PNTD readers.

Reviewer #3: (No Response)

**Results:**

-Does the analysis presented match the analysis plan?

-Are the results clearly and completely presented?

-Are the figures (Tables, Images) of sufficient quality for clarity?

Reviewer #1: Yes

Reviewer #2: Out of 443 patients, 23 are qRT-PCR positive. Adding NS1, IgM, and IgG data for those 23 patients is better.

Reviewer #3: (No Response)

**Conclusions:**

-Are the conclusions supported by the data presented?

-Are the limitations of analysis clearly described?

-Do the authors discuss how these data can be helpful to advance our understanding of the topic under study?

-Is public health relevance addressed?

Reviewer #1: (No Response)

Reviewer #2: The discussion part is not enough. It should expand and include a discussion of each result.

1.Why did the authors use the 31 years of age above and below as a seropositivity cut-off age in this study? Please explain it in the discussion part.

2.Are there any differences in clinical symptoms between patients with dengue fever alone, malaria alone, and dengue fever and malaria complications? If so, this would be beneficial information for clinicians. Please explain in the discussion section.

3.The positive rates for DENV, malaria, and mixed infections shown in Table 1 vary according to age distribution. Please discuss why this is the case in the discussion section.

4.Of the 443 patients, 220 tested negative for DENV and malaria. In the discussion section, please discuss the possibility of other infectious diseases in these patients.

Reviewer #3: (No Response)

**Editorial and Data Presentation Modifications?**

Reviewer #1: No problem

Reviewer #2: (No Response)

Reviewer #3: (No Response)

**Summary and General Comments:**

Reviewer #1: Akanbi et al. report on the prevalence of dengue virus infection in Yenagoa, Bayelsa State, Nigeria. They also investigated co-infection with dengue virus and malaria. Although Africa has been regarded as an endemic region for dengue virus, there are still relatively few studies on the subject. The authors collected specimens from patients with acute febrile illness in four hospitals in Yenagoa, and the study was well conducted. Additionally, they reported that dengue virus serotype 3 was the predominant type in the area during the study period in 2022. The authors suggest that some cases previously diagnosed as malaria may in fact have been dengue, which is an important observation.

Comments:

1.Table 1: The results for IgG-positive cases are missing, although this was mentioned in line 162.

2.References 4 and 5: These refer to studies conducted in specific areas. It would be helpful to also cite a reference that discusses global dengue prevalence.

3.Scientific writing: Some expressions are not appropriate for scientific reporting and should be revised. For example:

oLine 100: “(see Figure 1)” — consider rephrasing to integrate the figure reference more smoothly.

oLine 151: “we obtained ethical approval to …” — consider revising for a more formal tone.

Reviewer #2: The authors reported that 14.5% of febrile patients had acute dengue virus (DENV) infection, 42.4% had malaria, and 6.5% were coinfected with both in Bayelsa State, Nigeria, and that the findings highlight the need for routine DENV testing among patients with acute febrile illnesses in the region. The authors should consider the reviewers' comments. The author needs to write more clearly and accurately.

Reviewer #3: The manuscript PNTD-D-25-00583 submitted by Dayo Olufemi Akanbi is titled: The need for routine testing for dengue virus in Nigeria: detection and prevalence of dengue and coinfection with malaria among febrile patients in Yenagoa, Bayelsa State, Nigeria. Akanbi et al., used a cohort of febrile patients (N=443) collected in May-June 2022 from four hospitals to determine the seroprevalence of dengue infection, malaria and dengue/malaria co-infections. The authors performed rapid tests to detect DENV NS1, IgM and IgG, followed by multiplex DENV RT-PCR to serotype all NS1 and IgM positive samples. They identified for the first time in Yenagoa (City in Bayelsa State) the presence of acute and past dengue infections with some patients experiencing malaria coinfection. This study provides evidence that emphasize the need to implement routine dengue surveillance in Nigeria.

The authors addressed an important question regarding the seroprevalence of flaviviruses, particularly dengue, in Nigeria, where data remain limited. The manuscript is concise and generally well written; however, minor revisions are recommended to improve the clarity of the manuscript.

Minor comments:

The manuscript lacks references for the reagents used throughout the methodology section. The authors should include the sources and catalog numbers where appropriate to ensure reproducibility.

L125-129: It is unclear whether the method used for plasmodium parasite screening is a standard or validated procedure. The authors should clarify this and provide an appropriate reference to support the methodology.

Molecular assay details: The molecular assay section requires further detail. The authors should provide comprehensive information regarding the primers used, the targeted gene, the reaction conditions, thermocycling parameters. Even a brief description will help ensuring reproducibility.

Statistical analysis: The manuscript does not clearly describe the statistical methods employed. The authors should specify the statistical tests used to claim associations/correlations between parameters.

Summary. L77: The word “signs” does not fit the narrative. Maybe the word “evidences” will be more appropriated.

Title: The current title is long and may dilute the manuscript core findings. Consider shortening it to enhance clarity and focus. Suggested long title: Detection and Prevalence of Dengue and Coinfection with Malaria among Febrile Patients in Yenagoa, Bayelsa State, Nigeria. Suggested short title: Dengue Infection among Febrile Cases in Bayelsa State.

Additional detail is needed regarding how rural and urban areas were defined in this study. Since spatial risk for vector-borne diseases may not align with the administrative definitions of rural and urban areas, clarifying the characteristics of rural and urban areas in this study is important.

Figure 1: In the left panel of figure 1, it is unclear whether there is a between the Bayelsa state legend and the Nigeria boundary legend. If these are represented by different symbols or colors, the legend must be revised for clarity and visibility.

Study limitations and clinical data: The manuscript would benefit from a discussion of its limitations. Specifically, the absence of clinical data (e.g., WBC, liver enzyme levels, platelet counts etc…) limits interpretation of disease severity. Additionally, any analysis or discussion correlating clinical symptoms with coinfection status or dengue vs. malaria status would strengthen the manuscript.

**Figure resubmission:**

**Reproducibility:**



---

## [Decision Letter · Decision Letter 1]

24 Sep 2025

Dear Dr Akanbi,

We are pleased to inform you that your manuscript 'Detection of dengue, malaria, and additional causes of acute febrile illness: the need for expanded testing, Bayelsa State, Nigeria' has been provisionally accepted for publication in PLOS Neglected Tropical Diseases.

Best regards,

Ran Wang, M.D.

Academic Editor

Qu Cheng

Section Editor

Shaden Kamhawi

co-Editor-in-Chief

Paul Brindley

co-Editor-in-Chief

Reviewer #1:

Reviewer #3:

Reviewer's Responses to Questions

**Key Review Criteria Required for Acceptance?**

**Methods**

-Are the objectives of the study clearly articulated with a clear testable hypothesis stated?

-Is the study design appropriate to address the stated objectives?

-Is the population clearly described and appropriate for the hypothesis being tested?

-Is the sample size sufficient to ensure adequate power to address the hypothesis being tested?

-Were correct statistical analysis used to support conclusions?

-Are there concerns about ethical or regulatory requirements being met?

Reviewer #1: No problem after revision.

Reviewer #3: The authors addressed the comments by clarifying the method section including statistic tests used in the manuscript.

**Results**

-Does the analysis presented match the analysis plan?

-Are the results clearly and completely presented?

-Are the figures (Tables, Images) of sufficient quality for clarity?

Reviewer #1: No problem after revision.

Reviewer #3: The authors addressed the comments.

**Conclusions**

-Are the conclusions supported by the data presented?

-Are the limitations of analysis clearly described?

-Do the authors discuss how these data can be helpful to advance our understanding of the topic under study?

-Is public health relevance addressed?

Reviewer #1: appropriate

Reviewer #3: The authors addressed the comments and added the limitations of the study

**Editorial and Data Presentation Modifications?**

Reviewer #1: (No Response)

Reviewer #3: The figure 1 (top panel on the left) is still not clearly visible. The legend overlaps with the names on the map.

**Summary and General Comments**

Reviewer #1: (No Response)

Reviewer #3: The authors addressed most of the comments and improve the manuscript.

---

## [Editor Report · Acceptance letter]

Dear Dr Akanbi,

We are delighted to inform you that your manuscript, "Detection of dengue, malaria, and additional causes of acute febrile illness: the need for expanded testing, Bayelsa State, Nigeria," has been formally accepted for publication in PLOS Neglected Tropical Diseases.

Best regards,

Shaden Kamhawi

co-Editor-in-Chief

Paul Brindley

co-Editor-in-Chief
